# Assessing Geographic Overlap between Zero-Dose Diphtheria–Tetanus–Pertussis Vaccination Prevalence and Other Health Indicators

**DOI:** 10.3390/vaccines11040802

**Published:** 2023-04-05

**Authors:** Emily Haeuser, Jason Q. Nguyen, Sam Rolfe, Olivia Nesbit, Nancy Fullman, Jonathan F. Mosser

**Affiliations:** 1Institute for Health Metrics and Evaluation, University of Washington, Seattle, WA 98195, USA; 2Department of Global Health, School of Medicine and School of Public Health, University of Washington, Seattle, WA 98195, USA; 3Department of Health Metrics Sciences, School of Medicine, University of Washington, Seattle, WA 98195, USA

**Keywords:** immunization, spatial overlap, DTP vaccine, integrated service delivery, geospatial modeling, zero-dose children, vaccination, vaccine coverage, geographic inequality

## Abstract

The integration of immunization with other essential health services is among the strategic priorities of the Immunization Agenda 2030 and has the potential to improve the effectiveness, efficiency, and equity of health service delivery. This study aims to evaluate the degree of spatial overlap between the prevalence of children who have never received a dose of the diphtheria–tetanus–pertussis-containing vaccine (no-DTP) and other health-related indicators, to provide insight into the potential for joint geographic targeting of integrated service delivery efforts. Using geospatially modeled estimates of vaccine coverage and comparator indicators, we develop a framework to delineate and compare areas of high overlap across indicators, both within and between countries, and based upon both counts and prevalence. We derive summary metrics of spatial overlap to facilitate comparison between countries and indicators and over time. As an example, we apply this suite of analyses to five countries—Nigeria, Democratic Republic of the Congo (DRC), Indonesia, Ethiopia, and Angola—and five comparator indicators—children with stunting, under-5 mortality, children missing doses of oral rehydration therapy, prevalence of lymphatic filariasis, and insecticide-treated bed net coverage. Our results demonstrate substantial heterogeneity in the geographic overlap both within and between countries. These results provide a framework to assess the potential for joint geographic targeting of interventions, supporting efforts to ensure that all people, regardless of location, can benefit from vaccines and other essential health services.

## 1. Introduction

Since the inception of the Expanded Programme on Immunisation (EPI) in 1974 [1], global efforts to expand access to lifesaving vaccines have produced tremendous public health benefits, with an estimated 50 million deaths averted by vaccination activities between 2000 and 2019 alone [2]. Over the past four decades, country immunization programs have overseen large gains in coverage for vaccines included in the original EPI program, alongside the global rollout and scale-up of newer vaccines.

However, since 2010, these gains have stalled or reversed in many countries, and global vaccination coverage has largely plateaued [3,4]. In addition, disruptions to immunization delivery efforts due to the COVID-19 pandemic have resulted in additional, persistent declines in global vaccine coverage, with the coverage of key vaccines such as diphtheria–tetanus–pertussis (DTP) falling in many countries to the lowest levels in decades [5,6].

The stagnation and backsliding of global vaccine coverage in recent years emphasizes the need for new approaches to vaccine delivery. The Immunization Agenda 2030 (IA2030) aims to provide such a global strategy, coordinating and strengthening vaccination efforts around the world to ensure that “everyone, everywhere, at every age fully benefits from vaccines for good health and well-being” [7]. IA2030 also contains a strong strategic emphasis on the integration of vaccine delivery with other key health services [7], embedding immunization programs within the broader context of primary health care and global goals to achieve universal health coverage [8,9].

To achieve these ambitious goals, immunization programs must be equipped to reach “zero-dose” children—children who have never received a dose of a routine vaccine—including children and communities historically missed by immunization services. Operationally, “zero-dose” is generally proxied by “no-DTP”; that is, children who have never received a dose of a DTP-containing vaccine [10]. Recent work analyzing the complex paths from birth to full immunization in 92 countries emphasizes the importance of zero-dose children, as receipt of a first vaccine is strongly associated with additional vaccinations [11]. Furthermore, zero-dose children are more likely to have limited access to water, sanitation, and education [12] and live in poorer households [11]. A substantial number of zero-dose children also live in proximity to conflict [13]. Therefore, more deliberate provisioning of multiple interventions or services in contact with health systems or providers, including vaccination services, could be an efficient way to reach at-risk children and communities and reduce health inequalities.

To understand where and with which services integrated delivery could have the greatest impact for previously underserved communities, an understanding of the degree of overlap between no-DTP prevalence and other health gaps is needed. Numerous previous studies have assessed these relationships at an individual level, most commonly using data from household surveys [12,14,15,16]. At the population level, analyses of the spatial overlap between gaps in immunization coverage and other health services can complement these individual-level analyses. Spatial analyses conducted in recent years have emphasized the substantial degree of subnational inequality in vaccine coverage [13,17,18,19,20,21,22,23], as well as other key health services and indicators [22,24,25,26,27,28,29,30,31]. Fewer studies have assessed whether subnational distributions of zero-dose (or no-DTP) children are similar to those for other health indicators [32]. Some publicly available tools, such as the WHO Health Equity Assessment Toolkit [33], allow for powerful comparisons of health indicators within countries, although only for the years in which surveys have been conducted, and are limited to the geographic resolution of traditional survey methods (e.g., the first administrative level). Spatial overlap analyses can help to identify subnational areas and health services that may benefit most from integrated intervention.

Here, we propose a set of analyses that can be used to explore and quantify the degree of spatial overlap between populations of zero-dose children (proxied by no-DTP prevalence and counts) and gaps in vaccine coverage or other health-related indicators. Leveraging estimates of vaccination coverage from geospatial models and publicly available gridded estimates of other health indicators, we estimate patterns of spatial overlap in five example countries to demonstrate how these patterns may be explored both between and within countries, as well as over time. The approaches presented here can be expanded to other countries and health indicators and could serve as a resource when considering the possibility of joint intervention targeting.

## 2. Materials and Methods

### 2.1. Geospatial Estimation of Vaccination Coverage

For the purposes of this analysis, we used the prevalence of no-DTP (the proportion of children of the target age for vaccination who have not received any doses of a DTP-containing vaccine (DTPcv)) as a proxy for zero-dose children. We used a previously published geospatial modeling approach to estimate DTP vaccine coverage at the 5 × 5 km level [17], updating the approach to include more recent data and extending through 2019 (estimates were previously published for years 2000–2016).

To simplify the demonstration of these analyses, we selected the following five countries as examples for this analysis: Nigeria, Democratic Republic of the Congo (DRC), Indonesia, Ethiopia, and Angola. These countries were selected by ordering all the countries by the total number of estimated no-DTP children in 2019 [3], excluding countries for which spatial estimates were unavailable for two or more comparator indicators (Appendix A).

We searched the Global Health Data Exchange (GHDx) for household-based surveys containing information on DTP vaccination status between 2000 and 2019 [34]. We included surveys with information on DTP coverage information among children aged 12–59 months and excluded surveys that lacked subnational geographic information, had unrealistic coverage estimates, or contained areal data but were missing survey design variables that precluded the calculation of representative DTP coverage for each areal unit. From these five countries, we included data from 35 surveys with vaccination coverage information for 420,710 children from 11,047 GPS-located clusters and 2772 areal units. We calculated coverage at the most geographically granular level available for inclusion in the model. To better estimate the covariate effects and account for cross-border patterns of vaccine coverage, we modeled each country as part of a multi-country region (mirroring regions used to estimate MCV1 coverage by Sbarra et al. [18]), resulting in the inclusion of an additional 202 surveys including data for 1,200,877 children from other surrounding countries in the modeling process. A full list of included surveys can be found in Appendix A and excluded surveys (with rationale for exclusion) in Appendix A.

We defined DTP1 coverage as the proportion of children who have received at least one dose of a DTPcv. At the most granular geospatial resolution possible for each survey, we calculated DTP1 coverage for each birth cohort. We then used a previously described Bayesian continuation ratio ordinal regression model-based geostatistical estimation framework to estimate DTP1 coverage [17], aggregated these estimates to the second administrative level using population estimates from WorldPop [35,36] and a modified version of the Database of Global Administrative Areas (GADM) shapefile [37], and then calculated no-DTP prevalence as 1—DTP1 prevalence. For analyses using counts of zero-dose children, we similarly converted no-DTP prevalence to counts by multiplying the estimates of children under 1 year of age for each second-level administrative unit and year derived from the gridded estimates from WorldPop. For brevity and consistency throughout this manuscript, we refer to second-level administrative units as “districts” hereafter, while acknowledging that the nomenclature for these units varies between countries (e.g., local government areas in Nigeria). Additional details of the geospatial modeling strategy can be found in the Appendix A.

### 2.2. Spatial Estimates of Other Health Indicators

To assess the degree to which areas with a high prevalence or counts of zero-dose children may also exhibit gaps in other health services or outcomes, we identified and included the following five additional health indicators in our analyses: mortality among children under 5 years of age (U5M) [24], children with stunting [25], children with diarrhea who did not receive oral rehydration therapy (ORT) [38], prevalence of lymphatic filariasis (LF) [39], and individuals not sleeping with insecticide-treated bed nets (ITNs) [31]. These metrics were selected based on their persistent significance in the global health sphere, the role that subnational disparity plays in that persistence, and relationships to immunization that may give rise to potentially useful overlap analyses.

In addition to their significance in global health, these metrics were selected according to the availability of published estimates over time across multiple countries at a 5 × 5 km resolution. Estimates available in this format were most readily comparable to those produced for no-DTP prevalence. The estimation of these different metrics also employed geospatial modeling techniques that incorporated similar survey and other data sources and accounted for relationships with covariates, as well as correlations across space and time. The range of years with available estimates differed for each metric (Table 1). For each metric, we analyzed the overlap with no-DTP prevalence in the most recent year of data available, and for select analyses, we compared the overlap during the most recent year to that in the year 2000. Given their limited use in the country, missing ITN information was not available in Indonesia, but there was full coverage for all other indicators and countries.

### 2.3. Analyses of Spatial Overlap

For this analysis, we assessed the spatial overlap of no-DTP and these additional indicators in the context of assessing the degree to which the greatest burden for both no-DTP and the other indicators fell within the same districts. We fractionally aggregated the 5 × 5 km resolution pixel estimates for each metric to the same modified GADM shapefile [37]. Because prioritization decisions may be based not only on prevalence but also on total counts, we multiplied the respective prevalence estimates by the target population (Table 1) data available from WorldPop [35,36] to calculate the count estimates for each metric. For metrics with count data already available (i.e., for U5M, ORT, and LF), we used those values directly, although these were also based on WorldPop data. For both no-DTP and the health indicators, we assessed the overlap based on the mean estimates of prevalence or counts, without accounting for the uncertainty associated with all of these indicators.

In practice, decisions about prioritization for integrated service delivery are (and should be) made not only by considering the geographic patterns of the relevant indicators, but by accounting for a broad range of factors, including the available resources and data, and tailored by local expertise to each context [9]. For the purposes of this study, we used a highly simplified categorization scheme to illustrate the potential applications of spatial overlap analysis to contribute to the prioritization decisions. Similar analytic techniques, however, could easily be applied to other prioritization groupings. In this illustrative categorization approach, we assigned districts to population-weighted quartiles of burden for each metric, where districts with the highest values for each metric were in the top quartile and the districts with the lowest values for each metric were in the bottom quartile. Through population weighting, we ensured that the sum of target populations within each quartile were roughly equal. The scope of categorization needs and overlap assessment may vary between country-focused and global stakeholders. To explore the implications of these different frames of reference, we organized districts into quartiles both (1) within countries and (2) at a multi-national scale across countries. Similarly, we also categorized the districts into quartiles according to both (1) prevalence and (2) counts. To assess the full scope of overlap, we produced bivariate maps displaying overlap across all quartiles (Figure 1). We also produced simplified maps highlighting only those districts in the highest quartile for no-DTP, the respective comparator metric, or both. Finally, while we largely focused on comparing no-DTP categorization with each individual comparator metric, we also produced maps quantifying the number of metrics in the highest quartile in each district.

We also aimed to quantify the overall degree of overlap between no-DTP and the other health indicators using the summary metrics to facilitate high-level comparisons. We calculated the proportion of districts in the highest quartile for no-DTP that were also in the highest quartile for the other indicators. Furthermore, we devised an additional measure that was not reliant on the quartile categorization schema. We envisioned a scenario where vaccination stakeholders might prioritize districts by aiming to reach the greatest number of no-DTP children in the fewest districts possible. If these same districts were also targeted for simultaneous interventions for our comparator health indicators, what proportion of that country’s target groups for those indicators would be reached? We applied this hypothetical approach, serially targeting districts based on the number of no-DTP children, beginning with the targeting of the single district with the highest number of no-DTP children, then the two highest no-DTP districts, and so on. At each step, we calculated the cumulative proportion of individuals reached (for both no-DTP and the additional indicator), with each subsequent district targeted based upon the number of no-DTP children. By comparing these cumulative proportions between the two indicators for each set of serially targeted districts, we can calculate the area under the curve (AUC) to serve as a measure of overlap (Figure 2). This process is illustrated in step plots in Appendix A. As an example, an AUC of 0.5 indicates that geographic targeting based upon no-DTP reaches areas with equal proportions of no-DTP children and children with stunting. AUC values < 0.5 would indicate a smaller proportion of children with stunting reached, and AUC values > 0.5 would indicate greater proportions of children with stunting reached. We then analyzed AUC values between countries and indicators and over time, comparing AUC values in 2000 to those in the most recent year of available data.

### 2.4. Ethical Approval and Reporting Guidelines

Data were not obtained from subjects for the Global Burden of Diseases, Injuries, and Risk Factors Study or related analyses such as this study. Instead, we used pre-existing, publicly available, de-identified datasets that include, but are not limited to, administrative and survey-based vaccine coverage reports. Data were identified through online searches, outreach to institutions that hold relevant data such as ministries of health, or individual collaborator references and identification. Most of the data used are publicly available. Therefore, informed consent was not required. This study was approved by the University of Washington’s Human Subjects Division Study ID: STUDY00009060. Our study follows the Guidelines for Accurate and Transparent Health Estimates Reporting (GATHER; Appendix A).

## 3. Results

### 3.1. Mapping Overlap

#### 3.1.1. Country-Specific Overlap by Prevalence

Figure 3A shows an example bivariate map that illustrates the spatial overlap between the population-weighted quartile classifications for no-DTP and stunting in Nigeria, based on the prevalence of each indicator. In this example, when categorizing by prevalence within Nigeria, based on the available geospatial estimates for both no-DTP and stunting, higher-prevalence districts tended to be more widely distributed through the northern regions of the country, while the southern regions had a lower prevalence for both indicators. Overlap between no-DTP and stunting categorization was high; nearly two thirds of all districts in Nigeria (488 of 774 districts, or 63.0%) were designated to the same population-weighted quartile for both no-DTP and stunting. Figure 4A shows a simplified representation of the same analysis, restricting the mapped districts to only the high-quartile areas for each indicator. Of the 207 districts in the highest quartiles for either no-DTP or stunting, half of those districts (49.0%, or 100 of 207 total) were in the highest quartile for both indicators.

The spatial overlap between health indicators varies from indicator to indicator and country to country (Appendix A). In Ethiopia, for example, the locations with the highest no-DTP prevalence are located primarily in the east and south of the country (especially in the Afar and Somali regions) and are distinct from those with the lowest ITN coverage, which are located more centrally (for instance, in Amhara and Oromia) (Appendix A). In the Democratic Republic of the Congo, the geographic overlap between under-5 mortality and no-DTP prevalence is highly heterogeneous, with a mixture of high-prevalence areas for no-DTP, U5M, both, and neither indicator (Appendix A).

#### 3.1.2. Country-Specific Overlap by Counts

As expected, when these same analyses are repeated using an example categorization approach based on counts rather than prevalence, the results tend to emphasize areas of large populations—although this pattern is not universal across indicators and countries.

For the overlap between no-DTP and stunting in Nigeria, for example, when categorization is based on counts rather than prevalence, higher-quartile districts still tended to be in the northern regions of the country, while southerly districts tended to be in the lower quartiles (Figure 3B). Compared to the prevalence-based approach, there was more concordance between count-based classifications, with more than three fourths of all districts (597 of 774 districts, or 77.1%) being designated to the same quartiles for both no-DTP and stunting. Fewer districts were classified into the highest quartiles for either metric based on counts compared to prevalence (125 vs. 207 districts), but a greater proportion were in the highest quartile for no-DTP and stunting (78 of 125 districts, or 62.4%). There were 44 districts categorized in the highest quartile for both no-DTP and stunting according to both prevalence and counts (Figure 4).

However, these patterns again varied between countries and indicators (Appendix A). For Indonesia, for instance, locations that might be targeted for joint targeting based on spatial overlap between no-DTP and missed ORT would vary broadly depending on whether decisions were informed by analyses of prevalence or counts (Appendix A). In Angola, prevalence-based analysis of the overlap between no-DTP and ITN use identifies broad areas of the country that is potentially amenable to joint targeting (Appendix A). Due to the population distribution in the country, however, count-based analysis suggests that joint targeting opportunities might be focused upon relatively few locations (Appendix A).

#### 3.1.3. Overlap for Multiple Indicators

For some stakeholders, it may be of interest not only to understand the degree of geographic overlap between no-DTP and other health indicators individually, but also to identify locations that may be amenable to integrated intervention across many indicators. We, therefore, produced country-specific maps that show the number of health indicators in the highest quartile in each district, using our population-weighted classification approach. Here, we continue to show results from Nigeria as an example, although results for other countries can be found in Appendix A.

According to both prevalence and counts, more indicators were classified in the highest quartile in northern and northwestern Nigeria (Figure 5). Districts in southern Nigeria were largely only in the highest quartile for one to two indicators (missed ITNs and/or LF), whereas districts in northern and northwestern Nigeria had many cases of the overlapping classification for no-DTP, stunting, U5M, and missed ORT.

When classified by prevalence, high-quartile districts were relatively more concentrated across indicators in Nigeria compared to other countries (Figure 5A, Appendix A). More than two thirds of districts in Nigeria were categorized into the highest quartile for at least one of the six indicators analyzed (525 of 774, or 67.8%), but these proportions were even greater in all other countries, including 74.8% of districts in Indonesia (374 of 500), 81.1% of districts in DRC (194 of 239), 81.0% of districts in Ethiopia (64 of 79), and 87.7% of districts in Angola (143 of 163).

The opposite was true when categorizing the districts into population-weighted quartiles by counts. In this example, a much smaller proportion of districts—43.7%—were in the highest quartile for at least one indicator in Nigeria (338 of 774 districts; Figure 4B). This trend was consistent across other countries (Appendix A).

#### 3.1.4. Multinational Overlap by Prevalence

The analyses above focus on describing the spatial patterns of no-DTP and other indicators, based upon within-country classification for each indicator. For global or regional decision-makers, however, examination of the degree of spatial overlap across countries may be of interest. We, therefore, repeated these analyses, but instead categorized districts as those with the highest prevalence or counts for each indicator across all five example countries included in these analyses, rather than within the countries separately. Given the limited number of countries and indicators used in this analysis, these example results are meant to be illustrative only, to demonstrate the magnitude of differences in the perceived overlap when looking across rather than between countries and are not meant as policy recommendations.

Categorizing by prevalence across our five focal countries combined, population-weighted quartile assignments for no-DTP and stunting were markedly similar (Figure 6). The quartile classifications for no-DTP and stunting exactly matched (i.e., districts in the lowest quartile for no-DTP were also in the lowest quartile for stunting, etc.) in nearly half of all districts (795 out of 1755 total; 45.3%). Districts in the highest quartile across all countries for no-DTP could be found in every country, as well as districts in the highest quartile for stunting (Figure 7). The districts where the highest-quartile categorization for no-DTP and stunting overlapped largely fell within Nigeria and Angola, with 27.0% of districts in Nigeria and 50.3% of districts in Angola being in the highest category for both indicators (209 of 774 in Nigeria and 82 of 163 districts in Angola). While significant portions of DRC and Ethiopia were in the highest quartile for one indicator or the other, there was little overlap between indicators in these countries, and none in Indonesia (Figure 7).

Different patterns were observed for other comparator indicators (Appendix A). For instance, when comparing categorization for no-DTP and ORT across all five countries to that for no-DTP and stunting, fewer districts in Nigeria and Angola were in the highest quartile for both indicators, whereas larger areas of Ethiopia and DRC were in the highest quartile for both (Appendix A).

#### 3.1.5. Multinational Overlap by Counts

Categorization at the multinational scale was even more closely aligned between no-DTP and stunting when classifying districts according to counts rather than prevalence (Figure 8). When categorizing by counts, quartile assignment matched exactly between no-DTP and stunting in 71.1% of all districts (1248 of 1755 total). Far fewer districts were in the highest quartile when considered in terms of counts—only 5.8% of districts were in the highest quartile for either indicator using counts, compared to 35.5% of districts when considered by prevalence (102 vs. 623 out of 1755 districts, respectively; Figure 8 and Figure 9). In addition, the highest-quartile districts for either indicator fell largely in Ethiopia and DRC. The highest-quartile districts were scarce in the other three countries, making up <3% each for districts in Nigeria, Angola, and Indonesia. Only 49 districts were in the highest quartile for both no-DTP and stunting (2.7% of all districts), and more than half (28 of 49) were found in Ethiopia. This trend was largely consistent across all indicators (Appendix A).

### 3.2. Quantifying Spatial Overlap

District-level mapping, as in the analyses above, can help to identify subnational locations with potential for joint targeting. In some cases, however, it may be useful to quantify the degree of spatial overlap between no-DTP and another indicator in a single summary metric—i.e., to compare between countries or across comparator indicators. These summary metrics may help to determine the potential benefit of integrated services and delivery for some indicators compared to others, for instance.

#### 3.2.1. Percent Overlap of High-Quartile Districts

First, we identified all the districts in the highest quartile for no-DTP and calculated the proportion of those districts that were also categorized into the highest quartile for each of the other indicators. This proportion of overlap varied greatly between and within countries and indicators (Figure 10). The overlap was almost always higher when districts were classified based on counts rather than prevalence, with a few exceptions (e.g., overlap with LF or with ORT in several countries). For both prevalence- and count-based categorization approaches, the degree of overlap between no-DTP and other indicators tended to be lower in DRC compared to other countries; the proportion overlap was less than 50% for all comparator indicators except LF (where 66.2% of districts categorized in the highest quartile for no-DTP overlapped with LF highest-quartile categorization using prevalence, compared to 46.7% using counts).

Although the ranges between the indicators tended to be broad, there was nevertheless variation in consistency within most countries. For example, for categorization based on prevalence, there was some degree of overlap with no-DTP for every comparator indicator in Angola; the proportions of overlap ranged from 25.0% for LF to 62.8% for missed ORT. In Nigeria, on the other hand, proportions ranged from extremely low overlap with missed ITNs (0.6%) to high overlap with missed ORT (77.9%).

#### 3.2.2. AUC

In the more recent year of measurement, across countries and indicators, the median AUC was 0.43 (where AUC = 0.5 indicates equal proportions of the comparator indicator and no-DTP reached through no-DTP targeting, AUC < 0.5 indicates lower proportions of the population reached for the given indicator compared to no-DTP, and AUC > 0.5 indicates greater proportions of the population reached for the given indicator). The AUC for stunting in Nigeria was slightly above this value at 0.453 (Figure 2). The overall range of values for this measure was relatively narrow (Figure 11, Appendix A). Two-thirds of the observations fall between 0.39 and 0.46, with all indicators in Ethiopia and DRC falling within that range. The AUC was higher in Angola compared to other countries overall; only in Angola did any indicators reach an AUC > 0.5 (stunting at 0.52, LF at 0.55, and missed ITNs at 0.58), indicating even greater proportions of those target populations reached (compared to no-DTP populations reached). This finding is possible when the degree of geographic concentration is greater for other indicators than for no-DTP.

Based on AUC, across indicators, overlap with no-DTP was generally lower in 2000 compared to the more recent year measured in the countries included here, indicating broad reductions in spatial overlap over time (Figure 11). The largest decreases were for LF and missed ITNs in Nigeria, which were already lower than the other indicators in Nigeria in 2000 and these declined by 0.15 and 0.14, respectively. Angola was an exception to this trend, with a higher AUC in the more recent year across the indicators.

## 4. Discussion

In this study, we present a series of analyses of the distribution of no-DTP children and populations in need of other health interventions, using available subnational estimates of each indicator, and highlight their potential utility by applying these approaches to five example countries. These results demonstrate the substantial variation in joint geographic overlap between no-DTP and other health indicators, both between and within countries. In addition, the degree of spatial overlap and potential areas for joint geographic targeting vary depending on whether classification is based on prevalence or counts, and whether policy decisions are being made within or across countries. In general, the degree of spatial overlap between no-DTP and other indicators (measured by AUC) decreased over time for most comparisons and countries, with the exceptions of LF in Ethiopia and multiple indicators in Angola. For several of these analyses, we derived hypothetical categorization schemes for no-DTP children for illustrative purposes, such as population-weighted quartiles or serial targeting of districts based upon the estimated number of no-DTP children living in each district. We note, however, that these approaches could (and should) be tailored to reflect specific subnational prioritization plans under consideration in the future, while also expanding to include more countries and/or comparator indicators in the analysis. Taken together, the analytic approaches presented here form a foundation for future work to better understand the degree of geographic overlap between districts with high numbers of no-DTP children and those in need of other vital health services.

The comparator indicators presented here reflect a mixture of health service and health outcome measures, illustrating the different ways in which spatial overlap analyses might be applied. For instance, previous integration efforts have often included co-delivery of immunizations and ITNs [40], and areas with high LF prevalence and low immunization coverage may benefit from mass drug administration and immunization efforts. Reducing the disease burden of childhood diarrhea requires multifaceted approaches, such as preventive measures (including vaccination, i.e., for rotavirus) and access to treatment (including ORT) [41]. Malnutrition and immunization have complex interactions; malnourished children are at a higher risk for infectious disease mortality [42] and may benefit most from the protection of vaccines. Malnutrition may also affect immunologic responses to vaccination, and vaccination is an important component of multi-pronged interventions to reduce malnutrition [43]. Lastly, despite substantial progress, under-5 mortality in many countries is still significantly higher [44] than the stated Sustainable Development Goal (SDG) target of 25 or fewer deaths per 1000 live births by 2030 [8], and immunization is one of the cornerstones of efforts to reduce child mortality. Comparisons between gaps in vaccination coverage and these indicators, therefore, can illustrate a variety of potential uses for spatial overlap analyses.

For no-DTP children and communities that face barriers to accessing essential health services beyond immunization, integrating vaccine delivery with the delivery of other services could potentially provide substantial equity benefits. Integrated approaches also have the potential to increase the efficiency of health service delivery. As a result, integration has been a key theme of global immunization strategies over the past decades. The integration of immunization service delivery along with other public health interventions across one’s life course is one of the strategic priority goals of IA2030 [7], formed one of the strategic focus areas of the Global Immunization Vision and Strategy (2006–2015) [45], and was one of the guiding principles of the Global Vaccine Action Plan (2011–2020) [46]. The World Health Organization has also published extensive guidance for the integration of immunization services across one’s life course and within health systems [47].

Past efforts have focused on the integration of immunization services with other interventions in both campaign and routine immunization settings, including services such as ITN distribution, mass drug administration for deworming, vitamin A supplementation and nutritional services, family planning, HIV services, water and sanitation, and intermittent preventive therapy for malaria, among others [9,40]. Reviews of program experiences that implemented such integrated immunization activities suggest that integration can be challenging and highlight the need for a thoughtful consideration of the feasibility of joint intervention; careful, context-specific planning and implementation; strong community-based leadership; and timely and reliable monitoring strategies [9,48]. Analyses of the geographic overlap of populations in need of improved vaccination services and other interventions—such as those presented in this study—could serve as valuable additional input into this decision-making and planning process. Moreover, the heterogeneous patterns of overlap between countries and indicators illustrated by this study reinforce the need for context-specific decision-making about the integration of service delivery and integration plans that are tailored to the needs of each country and community.

This study is subject to several important limitations. First, this analysis focuses on district-level, population overlaps between the distribution of no-DTP children and other health services. This type of analysis helps to define geographic areas that might benefit from joint prioritization of immunization and other service delivery. This approach, however, does not examine other dimensions of overlap that may be important to understand when evaluating the potential benefits of integrated service delivery. These results should be paired with local expertise, as well as individual-level analyses such as those recently published [12], which can provide a more nuanced understanding of the associations between no-DTP status, lack of access to other health services, and other important non-geographic factors, such as poverty and race/ethnicity. Second, geospatial modeled estimates are often generated from survey data, which can vary in representativeness, temporal availability, and accuracy across indicators and between countries, and are subject to important forms of bias (including recall bias). Survey data representativeness may vary due to limitations of the available population estimates to inform sampling designs in some countries. In cases where populations at high risk for being zero-dose—for instance, those living in urban poor areas or migrant populations—are not adequately represented in the survey data, the resulting geospatial estimates will reflect these underlying biases. Third, these analyses rely on gridded population estimates from the WorldPop project [36] to convert between the prevalence of each indicator and counts of individuals at risk. In settings where no recent census data are available or migration is common, however, inaccurate population estimates could substantially bias prioritization decisions. To support accurate prioritization and planning, reliable target population estimates are critical. Last, we note that the classifications for the indicators presented here may not translate directly with the unmet needs. For example, coverage of ITNs on its own does not account for the endemicity of malaria. This limitation emphasizes the need for a framework such as that proposed here to be considered alongside a broad range of additional factors, context, and local expertise. For additional limitations, please see the Appendix A.

As this paper has highlighted, contextual knowledge is crucial for the effective use of any analyses to be used in decision-making. That contextual information can be highly localized and unique to each situation. We also note that the work in this paper is presented here without that contextual input of those most affected by under-immunization. While we have attempted to present many different analytical facets to address a range of possible use cases, we nevertheless acknowledge this critical component still missing from these analyses. Therefore, we invite feedback from global, regional, national and local experts in vaccine delivery and health service delivery as to how this work may be improved, modified and/or tailored to best support the efforts to reach zero-dose children and provide essential health services.

## 5. Conclusions

As the global immunization community works to fulfill the ambitious goals of IA2030, new strategies to reach zero-dose children and communities will be needed. Integrating immunization with other essential health services, as part of robust primary health care systems, has the potential to improve efficiency and achieve greater equity in health outcomes, particularly for communities that are most at risk. The potential benefits of integration—and the ideal strategies to plan and implement these efforts—are likely to vary from country to country. Spatial analyses of the overlap between gaps in immunization services and other key health indicators can help to define the potential for joint geographic targeting of integrated service delivery to help ensure a future where all people have equitable access to lifesaving vaccines.

## Figures and Tables

**Figure 1 vaccines-11-00802-f001:**
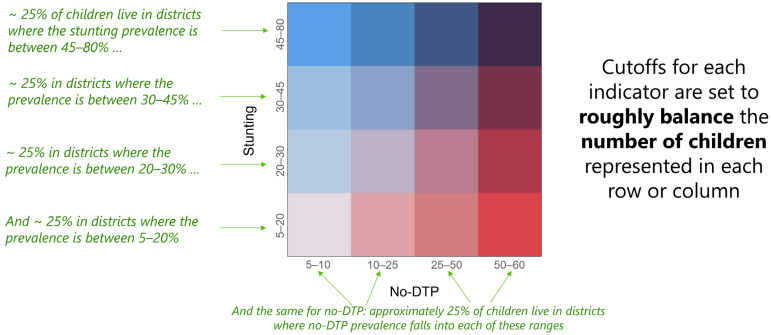
Example for a bivariate color legend used in maps to describe quartile classification overlap between no-DTP and the other health indicators (with stunting here as an example). For each of the two indicators, districts are distributed across four bins based on prevalence values such that the total target population value is roughly equal in each bin. Color bins along the diagonal (from bottom left to top right) indicate matching category assignments for the two indicators for a given district. Schema is used in figures representing all categorization quartiles.

**Figure 2 vaccines-11-00802-f002:**
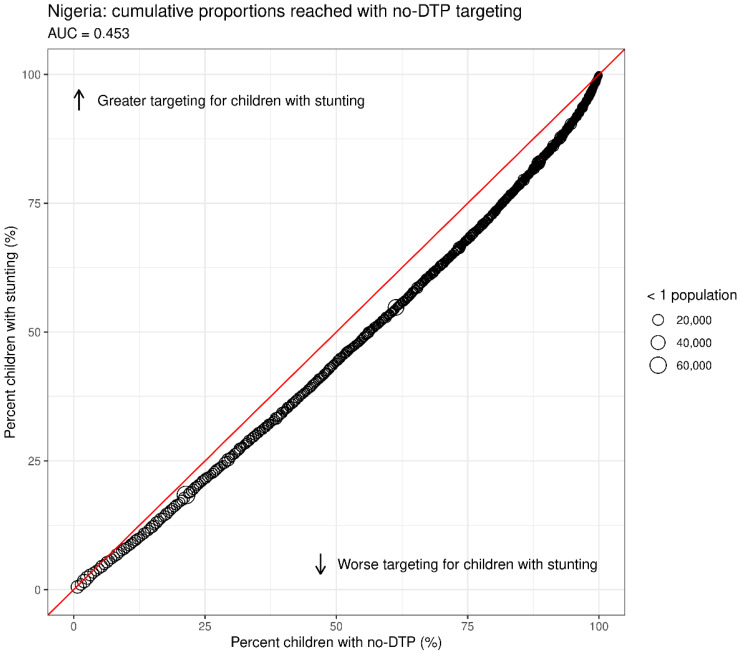
Area under the curve (AUC) scatter plot example. This example scatterplot visualizes how the area under the curve (AUC) can be used to quantify the proportion of children with stunting (Y axis) in Nigeria that could be reached through cumulative proportion targeting of districts for children with no-DTP (X axis). Individual points represent districts, ordered to begin with the district with the highest number of no-DTP children, then the second highest, until the cumulative proportion of no-DTP children reaches 100%. The red line represents AUC = 0.5, indicating equal proportions of children with stunting through cumulative no-DTP targeting. Curves below the red line are associated with AUC < 0.5 or smaller proportions of children with stunting, and curves above the red line are associated with AUC > 0.5, or greater proportions of children with stunting. Point size represents district population size of children under 1.

**Figure 3 vaccines-11-00802-f003:**
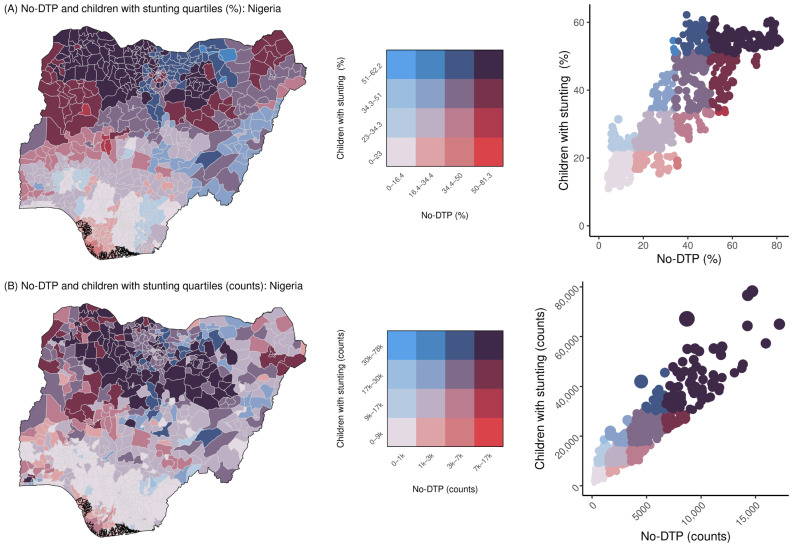
Country-specific Nigeria overlap for no-DTP and stunting, for all categorization quartiles. The top row (**A**) shows categorization based on prevalence, while the bottom row (**B**) shows categorization based on counts. Population-weighted quartile ranges for no-DTP and children with stunting are delineated in the bivariate color legends (center). District-level values are shown both as maps (left) and with scatterplots (right), with colors corresponding to quartile legend values. Point size in scatterplots reflects relative size of under-1 population in each district.

**Figure 4 vaccines-11-00802-f004:**
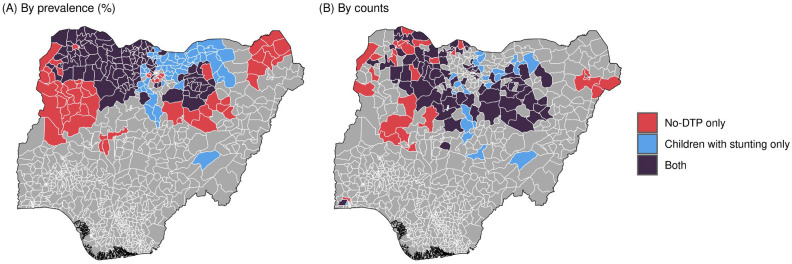
Country-specific overlap between no-DTP and stunting in Nigeria for highest category quartiles only. Districts in red are in the highest quartile for no-DTP only, blue are in the highest quartile for children with stunting only, and purple are in the highest quartile for both indicators. The map on the left (**A**) shows categorization based on prevalence, and the map on the right (**B**) shows categorization based on counts.

**Figure 5 vaccines-11-00802-f005:**
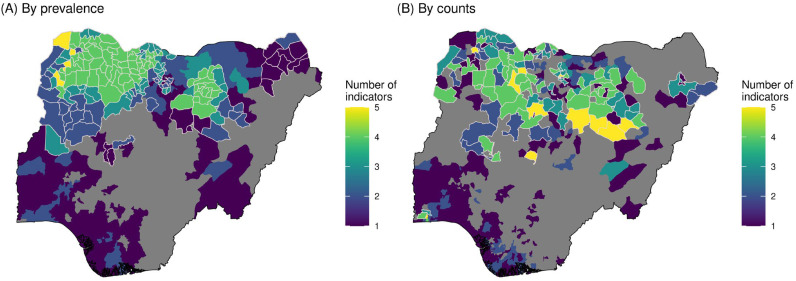
Country-specific multi-indicator overlap for Nigeria. Color given in each district reflects the number of indicators assigned to the highest quartile in that district. Districts outlined in white indicate those where no-DTP is among the indicators in the highest quartile. The map on the left (**A**) shows categorization based on prevalence, and the map on the right (**B**) shows categorization based on counts.

**Figure 6 vaccines-11-00802-f006:**
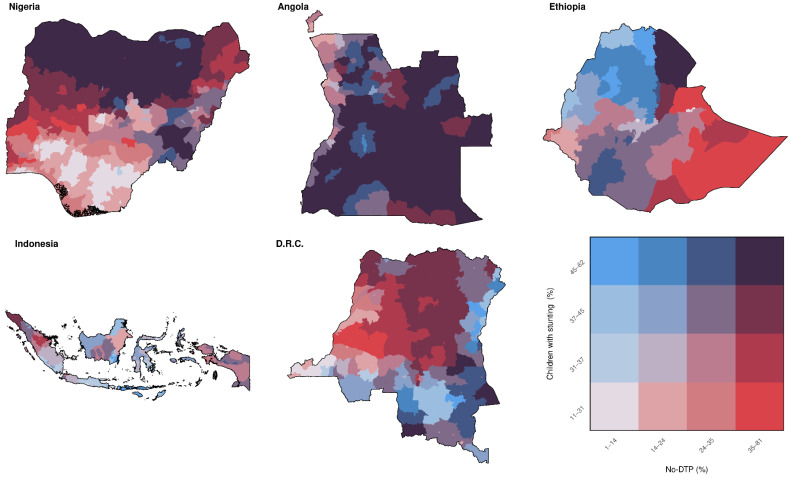
Multinational overlap between stunting and no-DTP for all categorization quartiles, based on prevalence. Ranges for population-weighted quartiles across the five example countries combined for no-DTP and children with stunting are delineated in the bivariate color legend (bottom right).

**Figure 7 vaccines-11-00802-f007:**
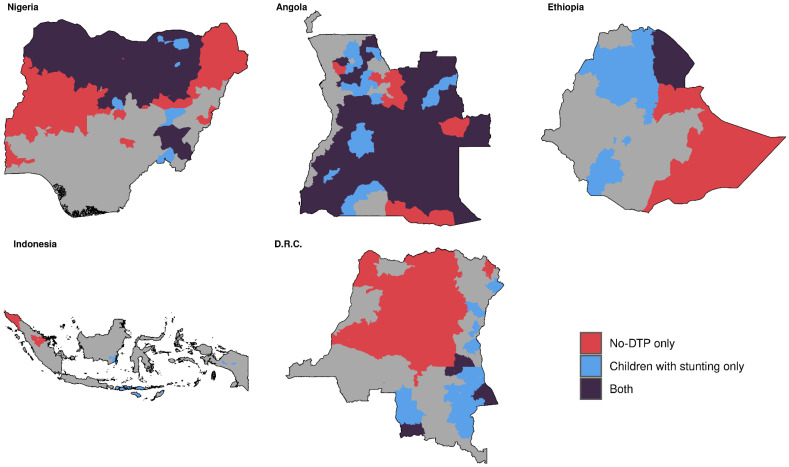
Multinational overlap between stunting and no-DTP for highest quartiles only, based on prevalence. Districts in red are in the highest quartile for no-DTP only, blue are in the highest quartile for children with stunting only, and purple are in the highest quartile for both indicators.

**Figure 8 vaccines-11-00802-f008:**
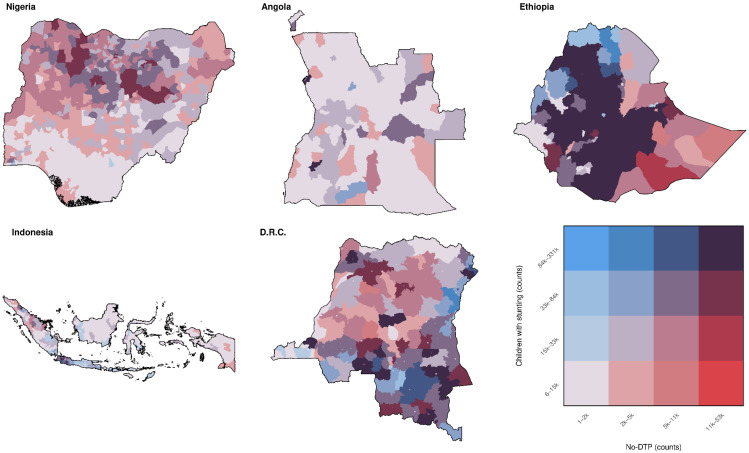
Multinational overlap between stunting and no-DTP for all categorization quartiles, based on counts. Ranges for population-weighted quartiles across the five example countries combined for no-DTP and children with stunting are delineated in the bivariate color legend (bottom right).

**Figure 9 vaccines-11-00802-f009:**
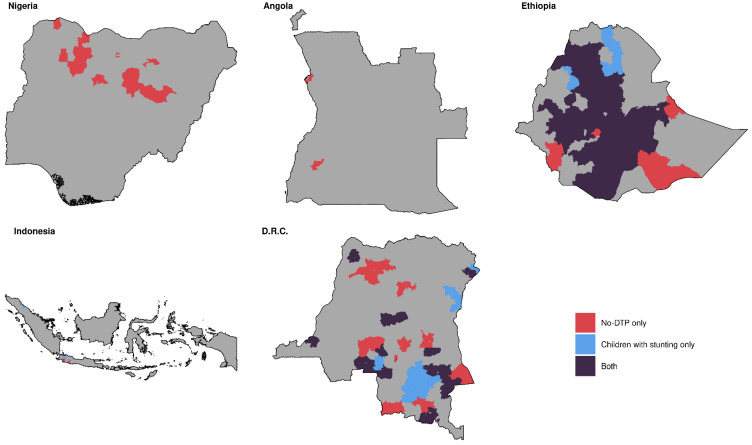
Multinational overlap between stunting and no-DTP for highest quartiles only, based on counts. Districts in red are in the highest quartile for no-DTP only, blue are in the highest quartile for children with stunting only, and purple are in the highest quartile for both indicators.

**Figure 10 vaccines-11-00802-f010:**
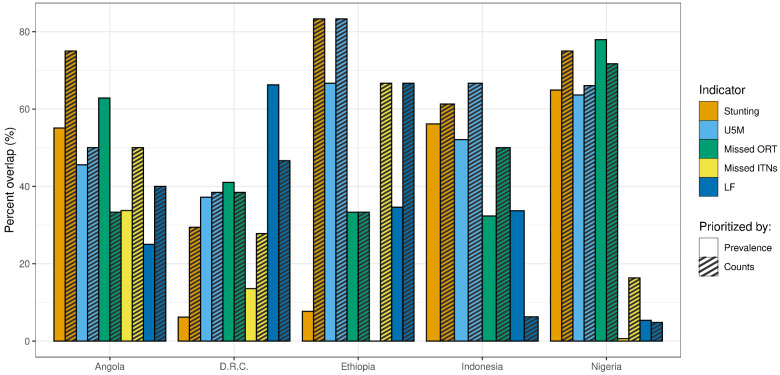
Overlap between districts in highest quartile categories for both no-DTP and comparator indicators, by country. Percent overlap indicates the proportion of districts in the highest quartile for no-DTP that are also in the highest quartile for the respective comparator indicators. Solid bars represent categorization based on prevalence, while striped bars represent categorization based on counts.

**Figure 11 vaccines-11-00802-f011:**
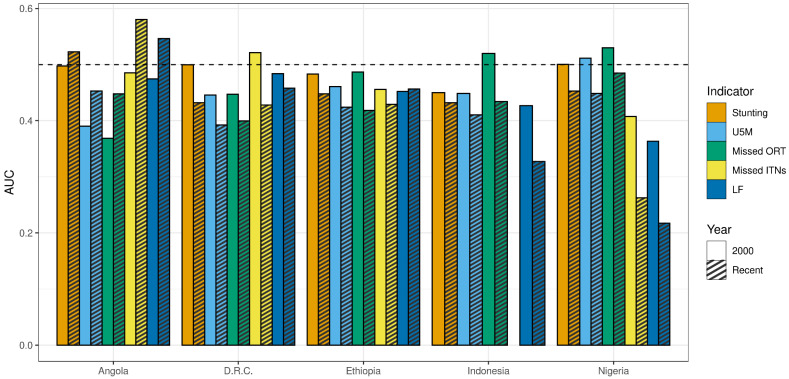
Comparison of AUC in 2000 and the most recent year of available data for each comparator indicator, by country. Solid bars represent values for the year 2000, while striped bars represent values for the most recent year of data available for the given metric (Table 1).

**Table 1 vaccines-11-00802-t001:** Details for health indicator estimates.

Indicator	Definition	Example Countries Available	Most Recent Year Available	Target Population Age Range	Citation
No-DTP	No-DTP prevalence rate	Angola, DRC, Ethiopia, Indonesia, Nigeria	2019	Under 1 year	Mosser, J.F. et al. [17] *
Stunting	Stunting prevalence rate	Angola, DRC, Ethiopia, Indonesia, Nigeria	2019	Under 5 years	Kinyoki, D.K. et al. [25]
U5M	Mortality probability and death counts	Angola, DRC, Ethiopia, Indonesia, Nigeria	2017	Under 5 years	Burstein, R. et al. [24]
Missed ORT	(1–ORT coverage) for children who had diarrhea	Angola, DRC, Ethiopia, Indonesia, Nigeria	2017	Under 5 years	Wiens, K.E. et al. [38]
Missed ITNs	(1–proportion of population that sleeps under an ITN)	Angola, DRC, Ethiopia, Nigeria	2019	All ages	Bertozzi-Villa, A. et al. [31]
LF	LF prevalence rate	Angola, DRC, Ethiopia, Indonesia, Nigeria	2018	All ages	Cromwell, E.A. et al. [39]

* Estimates updated to include additional years, geographies, and data sources.

## Data Availability

The findings of this study are supported by data available in public online repositories and data publicly available upon request of the data provider. Details of data sources and availability are publicly available in the Global Health Data Exchange (https://ghdx.healthdata.org/record/ihme-data/dtp-vaccines-zerodose-overlap). All computer code is available online and can be found at (https://github.com/ihmeuw/vaccines-zerodose-overlap).

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
