# Peer review of "Assessing Geographic Overlap between Zero-Dose Diphtheria–Tetanus–Pertussis Vaccination Prevalence and Other Health Indicators"

_vaccines, 2023, doi:10.3390/vaccines11040802_

Round 1

Reviewer 1 Report

Trying to read the manuscript I realize an error message occur all across the manuscript when figures are discussed - This makes the article  extremely hard to read and this problem must be corrected (see attachment).

Author Response

Comment: Trying to read the manuscript I realize an error message occur all across the manuscript when figures are discussed - This makes the article  extremely hard to read and this problem must be corrected (see attachment).

RESPONSE: We apologize for this inconvenience in interpreting this manuscript—the errors here appear to have been introduced in the uploading and re-formatting process for the journal. We have now corrected this error in cross referencing by hardcoding the Figure and Table references.

Reviewer 2 Report

This is a group with much experience in developing geospatial modelling for which they like to see applications in supporting healthcare development, especially in low-middle- and low-income countries. This is a fascinating domain to explore and the authors took a great initiative by trying to move to a more detailed level of administrative unit in a country presenting data about absence of one type of vaccination (DTP (prevalence and counting)) and making links to other health outcome measures (U5M, stunting, ORT, LF, and ITNs) that could be sensitive as well for aiming to create integrated healthcare development at local level. They developed an extensive report (110p) and from that report they are presenting in this article a sort of a summary how the model is working. In their article they refer to 5 countries to show that different results are seen in different countries (sic) but highly focus on one country, Nigeria.

I have two basic comments on this article. The article is very lengthy, and sometimes difficult to follow regarding the finality of the paper. If the objective is to demonstrate how the model is working based on the supplementary report they developed, I would propose to stick to that objective. In the introduction they can say that they develop that instrument as a complement to other tools available to support the ambition of a WHO-activity of IA2030. For explaining how the tool is working, they don’t need 5 countries, and they don’t need to explain the data they did not use, rather explain one country, Nigeria, with the data they used and the analyses done, + the results. Most interesting in that respect are the AUC-results (Figure 8 in the Supplement Report) because that help to indicate the link between No-DTP-level and the other outcome measures selected which has not been presented in the paper. The graphs in Figure 10 and 11 in the paper are very colourful but it took me a while to understand well what was presented. Maybe there is a better way to present the difference between prevalence and counts than by using now a different colour intensity. In the discussion section they can highlight that they did the exercise in different countries and that gave different results which is what is expected as no country is completely organised the same as other countries. Where I have difficulties in the discussion section, is to present 7 levels of limitations about the model tool which then question where is the benefit? We all know that working with models is an approximation of the reality, but bringing up all the limitations what models can do, is for someone who is unfamiliar with modelling, a reason for no pay and no believe in such a tool and therefore ignore its value. I would in the discussion section heavily highlight where the tool has advantages as compare to what is available (the HEAT-WHO-program), the historical context, the monitoring, etc. and limit the limitations to maximum 3 points which is being cautious about the precise interpretation of model-information, decision are never taken by modelling only, the selected outcome measures to make the link with healthcare development should be adjusted to local needs and demands.

This comes to my second basic comment which is that there is like no check/reference that happened with country authorities -or maybe it did happen but is not mentioned- whether what they developed is a demand and need that local authorities wanted to have. Somewhere the starting point of such a project should be, what can we do about the demand of local authorities to improve their policy. Here we have too much the impression that the researchers developed something very nice and colourful, but don’t know precisely how that will enhance a local policy and whether there is a demand for that info. The concrete application should help to give support the further implementation of the work done. If the request came from the countries that they need more granularity for better implementation of integrated healthcare development, that is a good introduction about why the program was developed. The impression given here is that there is a WHO program, a stagnation of vaccine implementation studied and observed, and the authors knew what is the next thing to do to change stagnation which is their tool. However, what would be most helpful is to know what we want to develop is a local demand that is most helpful. Maybe the latter did happen, but if that is the case that would be great to mentioned it under which circumstances it happened.

Some other minor points:

-repeated info in the introduction and the method section about country selection

-article 17 (line 96) should indicate the period in the text (2000-2016)…

-don’t mention the excluded surveys as you did not include them (line 106)

-not sure why to define DTP3 because it is not further used in the analysis (line 123)..

-when presenting the health indicators (line 135+) there are comments given that should be presented in the discussion section like the lines 155-160. This is an interpretation that should be in the discussion section.

-many error reference sources in the text

-line 167 had empty space..

-result section should add Figure 8 of Nigeria of the Supplemental report.

GADM (line 179): first mentioned in the text and not specified what it is

Reviewer 3 Report

The authors present a framework for analyzing the geographic overlap between children who have not received any doses of a DTP vaccine and five other indicators of inadequate health resources, some directly related to infectious disease (e.g., insecticide-treated bed nets) and others less so (e.g., stunting). This manuscript is well written, using accurate and precise language throughout. The discussion of study limitations is appropriate for the material presented. The authors provide multiple ways to interpret their data (e.g., examining quartile overlap using both prevalence and count data) so health officials can consider their specific resources and challenges when deciding how best to target any interventions. That the authors provided these options without advocating for one over was a refreshing acknowledgement of the diverse practical considerations faced in different countries and regions. The easy-to-interpret figures generated by Haeuser et al. will be a useful tool for policymakers looking to have the greatest impact on public health in a world of limited personnel and resources. I have only minor comments for the authors.

Minor Comments

1. There appears to be a typo resulting in missing numerical ranges in lines 167-168 and 181-182.

2. The formatting in Table 1 is somewhat awkward, presumably in an attempt to scale it to the page width for printing. Consider changing the citation column to reflect the appropriate citation numbers in your reference list rather than listing the citation in full within the table. The extra space will likely benefit the presentation of the first four columns.

3. It seems as though every place you attempt to refer to a Figure in the text instead has "Error! Reference source not found."

4. Figures 10 and 11 are somewhat difficult to interpret when viewed through a colorblind simulator. Consider using a different palette such as cbPalette or cbbPalette, and if appropriate use texture (e.g., diagonal lines) in place of the light and dark hues to designate the prioritization criteria and year.

5. Can you speculate in your discussion as to why the overlap between no-DTP status and the other health indicators has generally declined in more recent years?

Round 2

Reviewer 1 Report

The previous article problem has been corrected and moreover the authors improved the article according to the reviewers comments - it is not particularly exciting data but it deserves being published. Nice effort.

Reviewer 2 Report

Great work. Hope you are able to promote this tool in a best way to great local support.